# Why is patch size important? Dealing with context variability in segmentation networks

**Marc Balle Sanchez**[1,2]                                      SANCHEZ@IMFUSION.COM

**Samuel Joutard**[1]                                            JOUTARD@IMFUSION.COM

**Mohammad Farid Azampour**[2]                                   MF.AZAMPOUR@TUM.DE

**Raphael Prevost**[1]                                           PREVOST@IMFUSION.COM

[1] *ImFusion, Munich, Germany*

[2] *Computer Aided Medical Procedures & Augmented Reality, Technische Universität München*

**Editors:** Accepted for publication at MIDL 2025

## Abstract

Deep learning models have made significant advancements in medical image segmentation. Patch-based training is the standard practice for 3D segmentation models. During model deployment in hospital settings, resource constraints may require performing inference with reduced patch sizes. However, this might lead to a decrease in performance. In this study we demonstrate that patch size augmentation is a straightforward and effective approach to enhance the robustness of a 3D U-Net to different patch sizes during inference. Furthermore, we show that using a hypernetwork to adapt the U-Net to diverse patch sizes further enhances performance across the patch size spectrum.

**Keywords:** Hypernetworks, patch size augmentation

## 1. Introduction

Medical image segmentation plays a vital role in clinical workflows, enabling accurate diagnosis (Munir et al., 2019), treatment planning (Satheeskumar, 2025), and disease monitoring (Choy et al., 2023). Among existing architectures, U-Net (Ronneberger et al., 2015) has become one of the most widely adopted and effective models for medical image segmentation (Azad et al., 2024). Due to the high computational demands of processing 3D medical images, both training and inference commonly rely on patch-based strategies. The choice of patch size is inherently task-dependent and critically influences model performance. Specifically, the patch size used during training determines the amount of contextual information available to the model for learning the segmentation task. While larger patches offer a broader spatial context, they also entail greater computational cost. To enable efficient batching, a fixed patch size is typically used during training. Inference is often conducted using a sliding window approach with the same patch size as in training. However, in practical deployment, models might run in resource-constrained environments such as those found in hospitals. Reducing the patch size is a straightforward approach to alleviate computational demands. Nonetheless, this introduces a distribution shift in contextual information, which, as will be shown, can potentially degrade segmentation performance.

In this paper, we demonstrate the importance of these considerations and compare two potential solutions in the context of multi-organ segmentation. First, drawing inspiration from recent work (Beyer et al., 2023), we use patch size augmentation as a straightforward

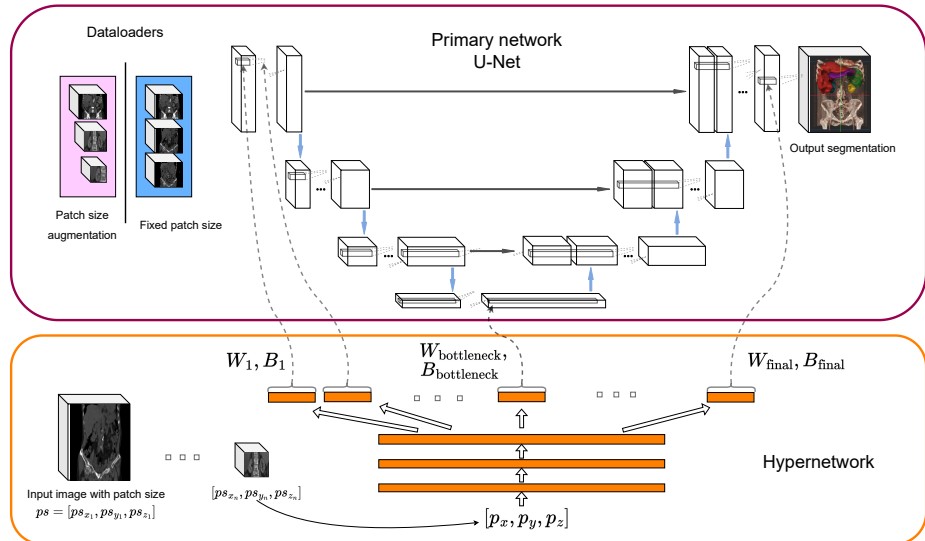

Figure 1: Considered models. The primary U-Net (red box) with the fixed patch size and augmented patch size dataloaders (blue and pink boxes) respectively correspond to U-Net FP and U-Net AP. The hyper-model (red and orange boxes combined) with the augmented patch size dataloader corresponds to HN.

approach extending the data domain to various patch sizes. Then, we explore the use of a hypernetwork (Ha et al., 2016) to condition the model on the patch size. The hypernetwork dynamically adapts the model to the input patch size by predicting a set of model weights which are optimal for this patch size. In medical applications, hypernetworks have shown success in conditioning registration networks on regularization strength (Hoopes et al., 2021; Mok and Chung, 2021) and segmentation networks on image spacing (Joutard et al., 2024), or target organ (Billot et al., 2024). In our case, the hypernetwork conditions the model on patch size by predicting optimal weights for a given amount of context.

## 2. Materials and Methods

**Dataset**  We conducted our experiments on the AMOS dataset (Ji et al., 2022), selecting the subset of 195 CT images in which all five following organs are segmented: liver, spleen, stomach, right and left kidneys. The dataset is randomly partitioned into 80% for training and 20% for evaluation. For testing the model, we used the Learn2Reg 2020 Task 3 dataset (Hering et al., 2022) with 30 images.

**Models**  In our experiments, we use a 3D U-Net with three levels, each consisting of four convolutional layers and 16 channels per layer in the first level. The hypernetwork is implemented as an MLP with three fully connected layers, each containing 50 units. We train the segmentation network in three different setups: using a fixed patch size of $[128]^3$(**U-Net FP**), augmenting the patch size (**U-Net AP**) and using a hypernetwork (**Hypernetwork U-Net (HN)**). These models are illustrated in figure 1.

| Model Configuration | Organs | | | | | Average |
|---|---|---|---|---|---|---|
| | Liver | Spleen | Stomach | R. Kidney | L. Kidney | |
| **HN** | 0.922* | 0.921* | 0.925 | 0.940* | 0.826 | 0.901* |
| **U-Net AP** | 0.886 | 0.882 | 0.915 | 0.936 | 0.755 | 0.897 |
| **U-Net FP** | 0.752 | 0.775 | 0.642 | 0.849 | 0.681 | 0.772 |

Table 1: Mean Dice scores over 500 different patch sizes. * denotes statistical significance for the comparison between HN and U-Net AP (Wilcoxon signed-rank test, $p < 0.01$).

## 3. Results and Discussion

Table 1 presents the Dice scores for each organ and model configuration on the test set. Inference is performed on 500 patch sizes, including 29 evenly spaced isotropic sizes from $[32]^3$ to $[256]^3$, and 471 randomly sampled sizes. Patch size augmentation and the hypernetwork both significantly improve performance ($\sim 13\%$ increase in Dice score) over fixed-size training, with the greatest gains seen for patch sizes smaller (see fig. 2) than U-Net FP's fixed training size ($[128]^3$), where loss of contextual information becomes critical. The results corresponding to isotropic patch sizes are presented separately in table 2, as these configurations are more prevalent in practical applications and therefore of greater relevance to end users. These findings suggest that patch size augmentation is a straightforward yet effective approach to improve the robustness of a model to varying patch sizes during inference. The hypernetwork, while requiring additional computational resources during training, delivers a significant improvement in performance over the U-Net PA, highlighting its potential for further enhancing model accuracy in diverse scenarios.

Furthermore, we observed that training a U-Net on small patch sizes (e.g., $[32]^3$ or $[64]^3$) resulted in instability and failure to converge, as this complex task requires larger contextual information. In contrast, only U-Net AP and HP were able to produce effective models for these smaller patch sizes, demonstrating that these configurations are essential for achieving a functional model under these conditions. This underscores the importance of the patch-size distribution extension for enabling the model to learn relevant features within the limited context regime.

## 4. Conclusion

In this study, we address the, often overlooked, distribution shift in contextual information during inference. We demonstrated the effectiveness of two approaches, one relying solely on patch size augmentation as a straightforward but effective solution, and a hypernetwork based approach that requires slightly higher training investment but yields optimal performances. We argue that both approaches could be seamlessly integrated into training pipelines for any model architecture, not just U-Net. We believe this work has the potential to facilitate model deployment in resource-limited scenarios such as hospitals. We will make the code publicly available at https://github.com/ImFusionGmbH/FlexPatchSize

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

## 5. Appendix

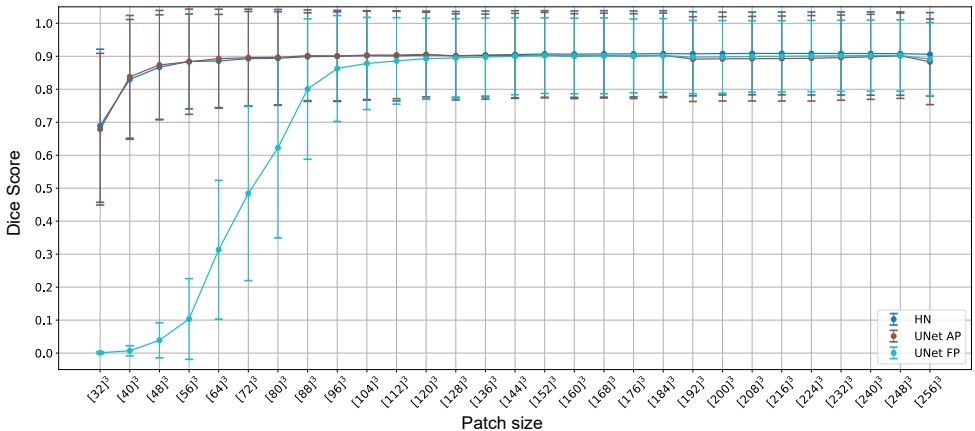

Figure 2: Dice score on the test set as a function of the patch size. Each point represents the mean over all organs and the error bars the standard deviation.

| Model Configuration | Organs | | | | | Average |
|:---:|:---:|:---:|:---:|:---:|:---:|:---:|
| | Liver | Spleen | Stomach | R. Kidney | L. Kidney | |
| **HN** | 0.904* | 0.914* | 0.912 | 0.937* | 0.826 | 0.892* |
| **U-Net AP** | 0.879 | 0.877 | 0.897 | 0.932 | 0.757 | 0.887 |
| **U-Net FP** | 0.724 | 0.734 | 0.668 | 0.779 | 0.665 | 0.731 |

Table 2: Mean Dice scores over 29 evenly spaced isotropic patch sizes from $[32]^3$ to $[256]^3$. * denotes statistical significance for the comparison between HN and U-Net AP (Wilcoxon signed-rank test, $p < 0.01$).

