# OpenReview forum: "Why is patch size important? Dealing with context variability in segmentation networks"
_MIDL.io/2025/Short_Papers — MIDL 2025 - Short Papers_

### Official Review · Reviewer_HrD1 · 2025-04-20

**Rating:** 5
**Confidence:** 5

**Summary:**

This paper explores two training strategies to mitigate the discrepancy in patch sizes between training and testing. The authors find that patch size augmentation alone effectively improves segmentation accuracy across different patch sizes. They also investigate a hypernetwork to generate weights when training with varying patch sizes, demonstrating the method’s effectiveness.

**Strengths:**

1. The paper addresses an often-overlooked technical problem arising from computational constraints.
2. It is well-prepared and presents solid results.
3. The proposed methods and baselines are well-justified.

**Weaknesses:**

1. It is unclear whether inference with varying patch sizes is practical. Ideally, one should select a fixed patch size based on test scenarios. A more convincing experiment would compare:
   - (1) Training only on 64×64×64 patches.
   - (2) Training on augmented patch sizes (e.g., 64×64×64 and 256×256×256) and testing on 64×64×64 under limited resources.
   If (2) outperforms (1), the argument would be stronger.

---

### Decision · Program_Chairs · 2025-05-01

Accept